

# Mucoactive agent use in adult UK Critical Care Units: a survey of health care professionals' perception, pharmacists' description of practice, and point prevalence of mucoactive use in invasively mechanically ventilated patients

Mark Borthwick[1], Danny McAuley[2], John Warburton[3], Rohan Anand[2], Judy Bradley[2], Bronwen Connolly[2,4], Bronagh Blackwood[2], Brenda O'Neill[5], Marc Chikhani[6], Paul Dark[7], Murali Shyamsundar[2] and MICCS collaborators—Critical Care Pharmacists

[1] Oxford University Hospitals NHS Foundation Trust, Oxford, United Kingdom
[2] Wellcome-Wolfson Institute for Experimental Medicine, School of Medicine, Dentistry and Biomedical Sciences, Queen's University Belfast, Belfast, United Kingdom
[3] University Hospitals Bristol NHS Foundation Trust, Bristol, United Kingdom
[4] Guy's and St Thomas' NHS Foundation Trust, London, United Kingdom
[5] Centre for Health and Rehabilitation Technologies, Institute of Nursing and Health Research, Ulster University, Newtownabbey, United Kingdom
[6] Anaesthesia and Critical Care, Division of Clinical Neuroscience, Nottingham University Hospitals NHS Trust, University of Nottingham, Nottingham, United Kingdom
[7] School of Biological Sciences, Salford Royal NHS Foundation Trust, University of Manchester, Manchester, United States of America

Corresponding author
Murali Shyamsundar,
murali.shyamsundar@qub.ac.uk

## ABSTRACT

**Background.** Mechanical ventilation for acute respiratory failure is one of the most common indications for admission to intensive care units (ICUs). Airway mucus clearance is impaired in these patients medication, impaired mucociliary motility, increased mucus production etc. and mucoactive agents have the potential to improve outcomes. However, studies to date have provided inconclusive results. Despite this uncertainty, mucoactives are used in adult ICUs, although the extent of use and perceptions about place in therapy are not known.

**Aims and Objectives.** We aim to describe the use of mucoactive agents in mechanically ventilated patients in UK adult critical care units. Specifically, our objectives are to describe clinicians perceptions about the use of mucoactive agents, understand the indications and anticipated benefits, and describe the prevalence and type of mucoactive agents in use.

**Methods.** We conducted three surveys. Firstly, a practitioner-level survey aimed at nurses, physiotherapists and doctors to elucidate individual practitioners perceptions about the use of mucoactive agents. Secondly, a critical care unit-level survey aimed at pharmacists to understand how these perceptions translate into practice. Thirdly, a point prevalence survey to describe the extent of prescribing and range of products

in use. The practitioner-level survey was disseminated through the UK Intensive Care Society for completion by a multi-professional membership. The unit-level and point prevalence surveys were disseminated cthrough the UK Clinical Pharmacy Association for completion by pharmacists.

**Results**. The individual practitioners survey ranked 'thick secretions' as the main reason for commencing mucoactive agents determined using clinical assessment. The highest ranked perceived benefit for patient centred outcomes was the duration of ventilation. Of these respondents, 79% stated that further research was important and 87% expressed support for a clinical trial. The unit-level survey found that mucoactive agents were used in 83% of units. The most highly ranked indication was again 'thick secretions' and the most highly ranked expected patient centred clinical benefit being improved gas exchange and reduced ventilation time. Only five critical care units provided guidelines to direct the use of mucoactive agents (4%). In the point prevalence survey, 411/993 (41%) of mechanically ventilated patients received at least one mucoactive agent. The most commonly administered mucoactives were inhaled sodium chloride 0.9% (235/993, 24%), systemic carbocisteine (161/993, 16%) and inhaled hypertonic sodium cloride (127/993, 13%).

**Conclusions**. Mucoactive agents are used extensively in mechanically ventilated adult patients in UK ICUs to manage 'thick secretions', with a key aim to reduce the duration of ventilation. There is widespread support for clinical trials to determine the optimal use of mucoactive agent therapy in this patient population.

# INTRODUCTION

Patients admitted to critical care frequently have acute respiratory failure and often require ventilation (*Narendra et al., 2017*; *Vincent et al., 2002*). Acute respiratory failure (ARF) may be due to a neuromuscular issue, secondary to an acute exacerbation of chronic obstructive pulmonary disease (COPD), an alveolar process (e.g. cardiogenic and noncardiogenic pulmonary oedema and pneumonia) or a vascular disease such as pulmonary embolism (*MacSweeney, McAuley & Matthay, 2011*). Large numbers of patients receive respiratory support in ICUs with an estimated 116,000 adult admissions in the UK alone for respiratory support every year (*Harrison, 2014*).

Ventilatory support and the requisite associated medications such as analgesics and sedatives are essential medical interventions that can affect other physiological mechanisms and may worsen them (*Jelic, Cunningham & Factor, 2008*). Mucus production and rheology is altered, and when combined with the diminished effectiveness of cough reflexes, impaired mucociliary clearance and the influence of gravity and body positioning on mucus flow, lead to mucus accumulation and plugging. The accumulation of mucus may also facilitate the growth of microorganisms, leading to infection (*Mietto et al., 2013*; *Kalanuria, Zai & Mirski, 2014*; *Icard & Rubio, 2017*; *Konrad et al., 1994*; *Dickson, 2016*).

The role of mucoactive agents is established in various chronic respiratory conditions such as COPD and cystic fibrosis where several therapeutic agents are used as aids to mucus clearance. These agents are administered either topically (inhaled/nebulised/intratracheal instillation) such as hypertonic sodium chloride solutions and recombinant deoxyribonuclease, or systemically such as carbocisteine (*Tarrant et al., 2017*; *Wark & McDonald, 2018*; *Yang & Montgomery, 2018*; *Cazzola et al., 2017*). The available literature regarding mucoactive agent use in critically ill patients with acute respiratory failure suggests widespread but inconsistent practice in the context of limited evidence (*Jelic, Cunningham & Factor, 2008*; *Papacostas, Luckett & Hupp, 2017*; *Van Meenen et al., 2018*; *Anand et al., 2019*; *Icard & Rubio, 2017*).

There are no published data on the use of mucoactives in UK critical care units. In order to gain as comprehensive an insight as possible, we conducted three surveys to explore various aspects of UK practice:

Survey 1 aimed to characterise how individual practitioners view the use of mucoactive agents (practitioner-level survey);

Survey 2 aimed to explore which mucoactive agents are used across adult UK critical care units and discover the extent of guideline use (unit-level survey);

Survey 3 aimed to determine the actual usage pattern of various mucoactive agents currently in clinical use (point prevalence survey).

## MATERIALS & METHODS

Survey questions were developed and prepared in Survey Monkey (Survey Monkey Inc, San Mateo, California, USA) for the practitioner-level and unit-level surveys. A data capture tool for the point prevalence survey was developed in Excel (Microsoft Excel 2010, Redmond, WA, USA). All survey instruments were piloted and iteratively refined.

The practitioner-level survey was distributed to the membership of the UK Intensive Care Society (ICS) via a newsletter on 17 September 2018. Responses were collected until 8th January 2019. Reasons for mucoactive agent use were collected by asking respondents to rank a list of clinical conditions and indications (see supplementary appendix).

Due to familiarity with medicines practice amongst multiple prescribers across critical care environments, pharmacists were targeted as respondents of choice to report unit-level and point prevalence surveys (*Intensive Care Society & Faculty of Intensive Care Medicine, 2019*).

For the unit-level survey, respondents were asked to rank unit practice regarding indications and anticipated benefits of mucoactive agents, to specify which agents are in use, and to provide copies of any guidelines used. Invitations to participate, including the survey link, were distributed electronically to UK critical care pharmacists via the UK Clinical Pharmacy Association. Responses were collected from 4th April 2018 to 25th April 2018.

For the point prevalence survey, the Excel tool was circulated electronically to all the pharmacists via the UK Clinical Pharmacy Association. Participants were requested to collect data for one 24 h period of their choosing for each critical care unit between the

18th March 2019 and 23rd April 2019. Additional forward distribution to further critical care pharmacy contacts was requested. To encourage survey participation, reminders that including response rate statistics were circulated at regular intervals via email or Twitter until the surveys closed. Only one survey response for each unit was allowed.

The survey questionnaires are attached as an online supplement.

The denominator for UK critical care units was defined as the number of units participating in the Intensive Care National Audit and Research Centre Case Mix Programme (England, Wales, Northern Ireland, $n = 276$) and Scottish Intensive Care Society Audit Group ($n = 63$).

Response data from Survey Monkey were downloaded to Excel (Microsoft Office, WA, USA) for analysis. Ranked data were converted into Borda counts for comparison. Briefly, this method assigns a score to each choice made, depending on the ranked position. The total score for each category is calculated by summation of these individual scores. For example, if there are three choices to rank in a category, each ranked first choice attracts a score of 3, each ranked second choice attracts a score of 2 and each ranked third choice attracts a score of 1. All scores are totaled for each category. The higher the score, the higher the choice is favoured (*Emerson, 2013*).

Some results of the unit-level survey have been published elsewhere as an abstract (*Borthwick et al., 2019*).

## RESULTS

### Practitioner-level survey

Questionnaires were sent to 3,099 clinician members of the ICS, of whom 225 completed the survey. The response rate was 8% but represented 37% of UK adult ICUs. Medical consultants formed the majority of the respondents and followed by medical trainees (Table 1). The highest proportion of respondents were from a 'mixed medical/surgical ICUs (including trauma)' (86%) followed by 'other' (8%). The most highly ranked clinical indication was 'thick secretions', followed by 'COPD' (Fig. 1). Except for one respondent, the assessment of 'thick secretions' was always based on clinical observation. Improved sputum clearance was the most highly ranked perceived benefit from mucoactive use (78%) while reduced time of mechanical ventilation (14%) was the most highly ranked patient centred clinical outcome (Fig. 2). A significant majority (79%) of respondents indicated that further research in the use of mucoactive agents was important and a similarly high percentage (87%) of respondents expressed their support for a clinical trial. However, a high proportion of respondents (28%) indicated they would not support a clinical trial that does not allow topical 0.9% sodium chloride as part of standard care.

### Unit-level survey

Pharmacists representing 128/341 critical care units responded (38%; including two additional units that do not provide data for the ICNARC case mix programme). Mucoactives were used in 106/128 ICUs (83%). The most highly ranked indication for mucoactive use was 'thick secretions' (Fig. 3). 'Reduced ventilation time' was the most highly ranked expected patient centred clinical outcome, with improvement in gas exchange

**Table 1 Practitioner level survey—breakdown based on professional group.**

| Professional group | Number of responses (%[a]) |
| --- | --- |
| Medical –Consultant | 178 (80) |
| Medical –Trainee | 22 (10) |
| Medical –Other | 2 (<1) |
| Physiotherapist | 11 (5) |
| Nursing | 11 (5) |
| Not available | 1 (<1) |

Notes.
[a]% is rounded to the nearest whole number.

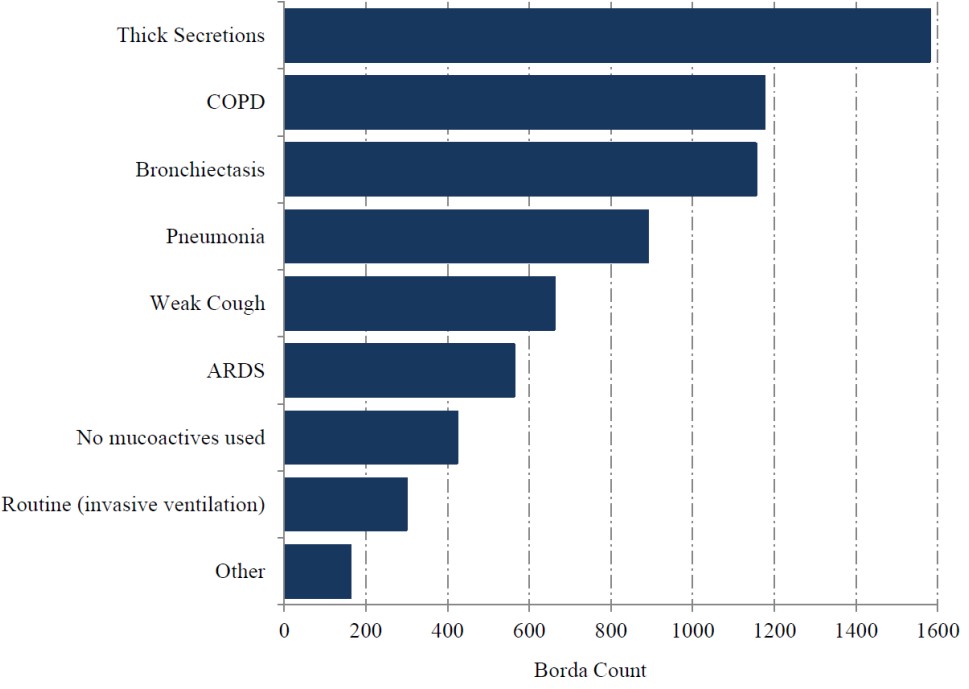

**Figure 1 Usual indication for mucoactive agents in use.**

also identified as an important benefit (Fig. 4). The highest ranking mucoactive agents reportedly in use were topical isotonic saline (ITS) and systemic carbocisteine respectively (Fig. 5). A wide range of topical hypertonic saline solutions and two concentrations of topical N-acetylcysteine were also reported (Fig. 5). Only five ICUs provided local guidelines directing the use of mucoactive agents (4%).

## Point prevalence survey

Critical care pharmacists representing 63% of UK critical care units contributed to the survey and included data from 993 invasively mechanically ventilated patients. 41% of patients were prescribed one or more mucoactive agents, confirming the wide prevalence of mucoactive use in mechanically ventilated patients. Excluding isotonic saline, the prevalence of mucoactive agents in invasively mechanically ventilated patients was 27.1%.

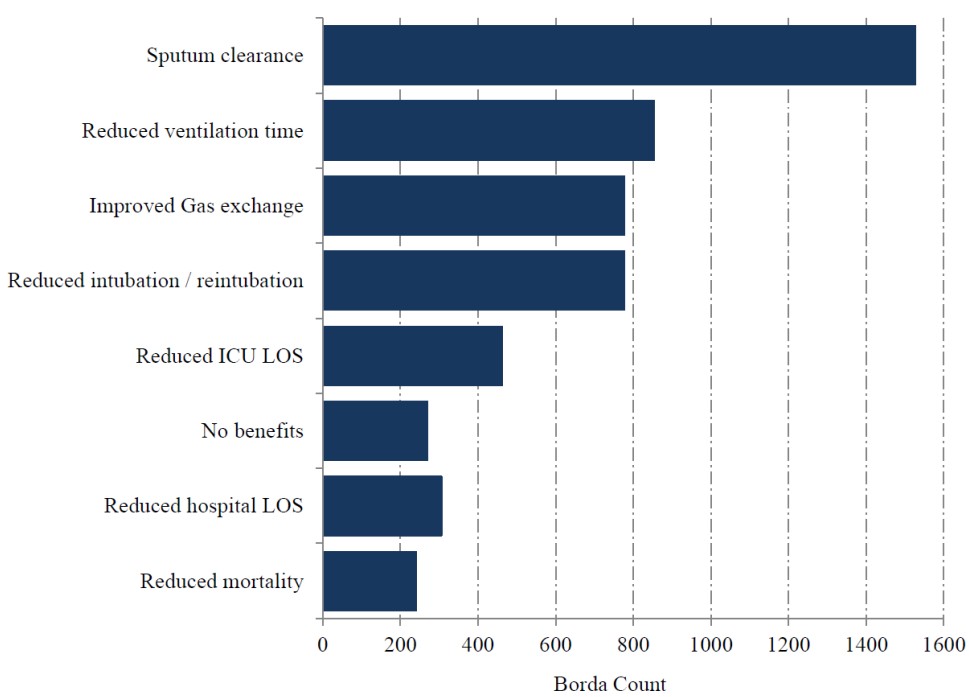

**Figure 2  Expected clinical benefit for mucoactive agents in use.**

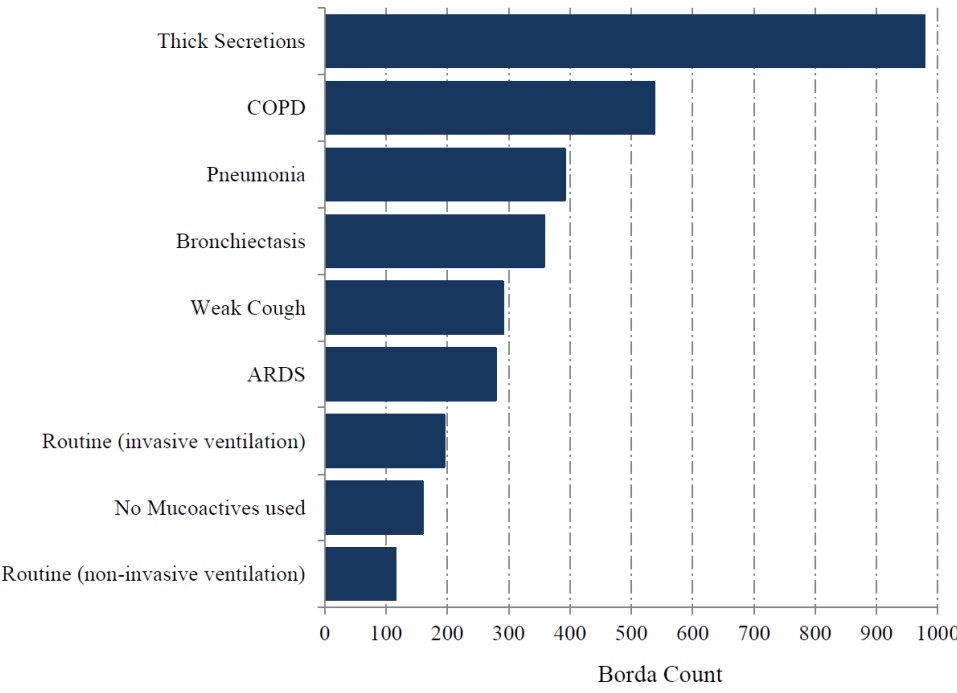

**Figure 3  Usual indication for mucoactive agents in use.**

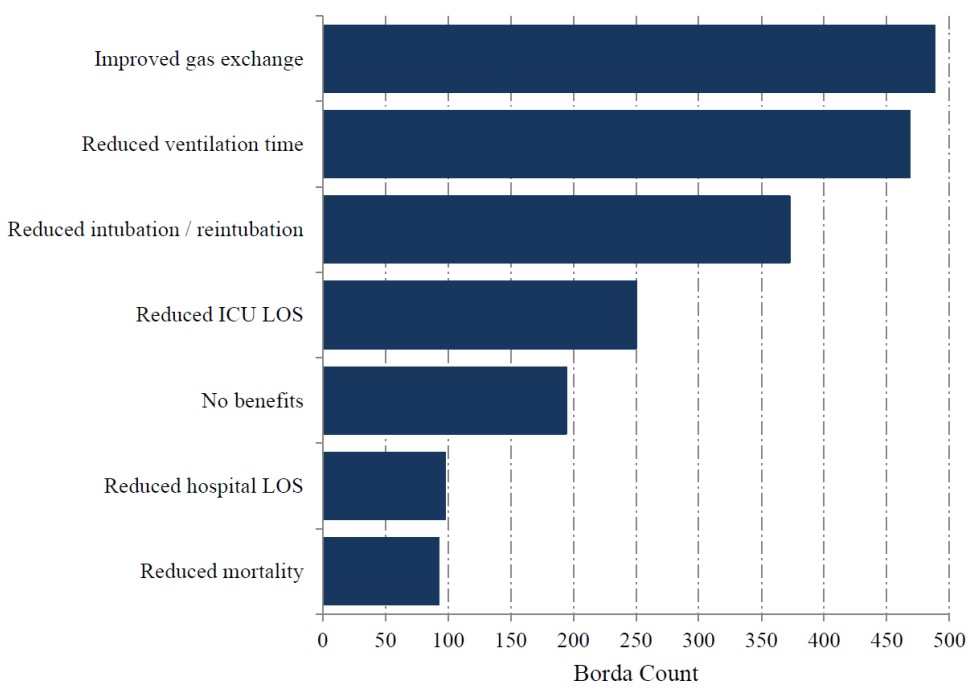

Figure 4 Expected clinical benefit for mucoactive agents in use.

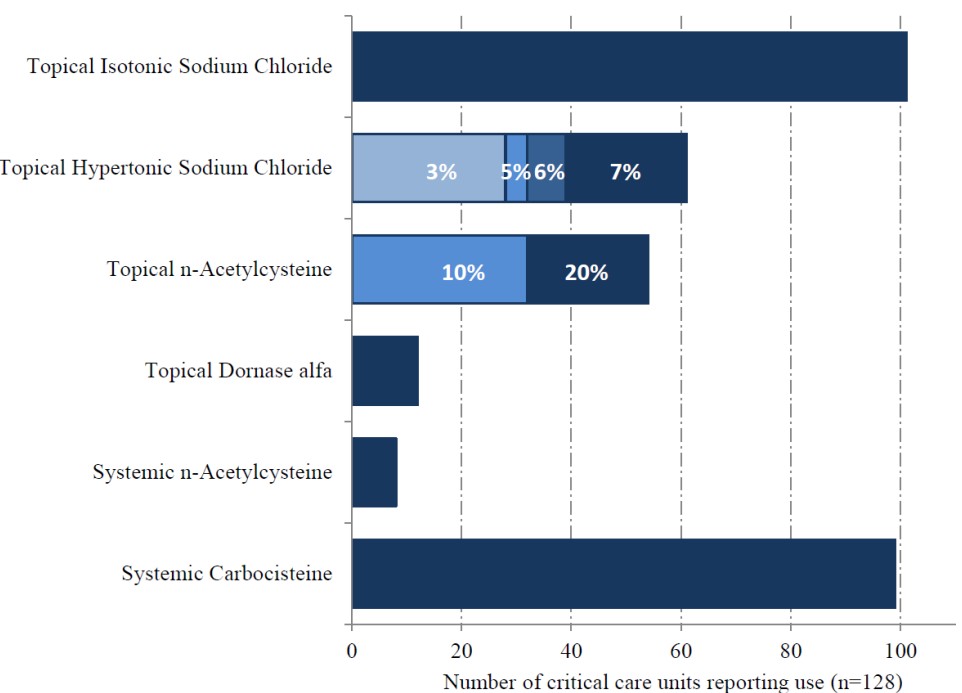

Figure 5 Topical and systemic mucoactive agents in use.

**Table 2** Prevalence and types of mucoactive agents used in UK ICUs—point prevalence survey.

|  | Yes (%) | No (%) |
|---|---|---|
| Any mucoactive use | 411 (41.4) | 582 (58.6) |
| Nebulised/intra-tracheal 0.9% saline | 235 (23.7) | 758 (76.3) |
| Systemic carbocisteine | 161 (16.2) | 832 (83.8) |
| Nebulised hypertonic saline | 127 (12.8) | 866 (87.2) |
| Nebulised/intra-tracheal N-acetylcysteine | 38 (3.8) | 955 (96.2) |
| Nebulised DNase | 1 (0.001) | 992 (99.999) |

Nebulised or intra-tracheal isotonic saline, systemic carbocisteine and hypertonic saline were the three most frequently used mucoactive agents in invasively mechanically ventilated patients (Table 2). The most frequent concentration range for hypertonic saline was 5–7%.

## DISCUSSION

This study provides the first detailed description of mucoactive use in mechanically ventilated patients within critical care in the UK. There are no comparable studies describing the perceptions and prevalence of mucoactive use in mechanically ventilated ICU patients globally, hence our study addresses this important knowledge gap.

The unit-level survey shows that mucoactive agents are used in the majority of critical care units (83%), and the point prevalence survey finds that a high prevalence amongst mechanically ventilated patients (41%). Given the response rate from the unit level and point prevalence surveys of up to two-thirds of respondents (38% and 63%), we believe the results to be generalizable to UK critical care practice.

Mucoactive agents act through a variety of mechanisms, such as decreasing viscosity, increasing mucus water content and stimulating cough. They may have the potential to improve patient outcomes in mechanically ventilated patients and could reduce the risk of secondary bacterial infection (*Mietto et al., 2013*; *Balsamo, Lanata & Egan, 2010*). Some mucoactive agents may also have a direct anti-inflammatory effect (*Balsamo, Lanata & Egan, 2010*). Notwithstanding a plausible biological rationale, the evidence for clinically meaningful beneficial endpoints is conflicting. Recent systematic reviews investigating the role of mucoactive agents in acute respiratory conditions including mechanically ventilated patients highlight both the lack of evidence and the low quality of current evidence to support clinical practice (*Tarrant et al., 2017*; *Papacostas, Luckett & Hupp, 2017*; *Anand et al., 2019*; *Icard & Rubio, 2017*). There was no consistent benefit on various clinical outcomes including duration of mechanical ventilation, duration of ICU stay, mucus clearance, radiological changes and oxygenation while there is some evidence of harm (*Wong, Anderson & Shyamsundar, 2019*). A major limitation of the available data is that the studies are small and inadequately powered.

A recent large multi-centre trial of patients undergoing mechanical ventilation has found 'on demand' N-acetylcysteine nebules for thick secretions to be non-inferior to routine use (*Van Meenen et al., 2018*), but this study had no placebo arm and so any overall benefit of N-acetylcysteine remains unclear.

Despite a lack of robust supporting evidence, our study confirms a high prevalence of mucoactive agent use, with the most common indication for 'thick secretions' and a clinical aim of reducing duration of ventilation. Our study also confirms the high prevalence of the use of 0.9% sodium chloride through the intratracheal route while the other common mucoactive agents in use are systemic carbocisteine and topical hypertonic saline. The clinical benefit of topical mucoactive agents such as nebulised isotonic saline that exert their action through simple airway humidification is unlikely to be significant in mechanically ventilated patients though use is common. Future clinical trials of mucoactive agents should include 0.9% sodium chloride as a component of standard care due to the very wide prevalence of use, and opposition to any study that does not allow this as part of standard care.

The strengths of this study are the inclusion of three different surveys that enable us to understand healthcare professionals' perception of the indication and benefits of mucoactive agents, as well as objective evidence of their actual pattern of use. For the first time, this study clearly establishes the type and range of mucoactive agents used in critical care in the UK. The response rate for the point prevalence study was more than 60% and is therefore likely to provide an accurate reflection of clinical practice. The limitations of the study are that the actual number of respondents for the practitioner-level survey was low which may limit the generalisability of this aspect, and that the point prevalence survey was conducted during spring when there could be seasonal variation in the use of mucoactive agents.

## CONCLUSIONS

This study finds that there is a high prevalence of mucoactive agents use in mechanically ventilated adult patients in the UK in spite of the lack of robust evidence of benefit. The highest ranked indication to initiate mucoactive agents is thick secretions with duration of ventilation the patient centred outcome ranked highest for perceived benefit in the practitioner level and highly in the unit-level survey. There is a need for a well-designed, adequately powered multicenter trial of commonly used mucoactive agents in patients that are mechanically ventilated and have thick secretions. There is widespread support for undertaking such a trial.

## ACKNOWLEDGEMENTS

The authors acknowledge the support of the Intensive Care Society and the UK Clinical Pharmacy Association for their support in disseminating the surveys.

### Funding

The authors received no funding for this work.

### Competing Interests

The authors declare there are no competing interests.

## Author Contributions

- Mark Borthwick and Murali Shyamsundar conceived and designed the experiments, performed the experiments, analyzed the data, prepared figures and/or tables, authored or reviewed drafts of the paper, and approved the final draft.
- Danny McAuley, John Warburton, Rohan Anand, Judy Bradley, Bronwen Connolly, Bronagh Blackwood, Brenda O'Neill, Marc Chikhani and Paul Dark conceived and designed the experiments, performed the experiments, analyzed the data, authored or reviewed drafts of the paper, and approved the final draft.

## Data Availability

Data is available at Queen's University Belfast: Shyamsundar, M. (Creator) (30 Aug 2019). Mucoactive agent use in adult UK Critical Care Units: a survey of health care professionals perception, pharmacists' description of practice, and point prevalence of mucoactive use in invasively mechanically ventilated patients . Queen's University Belfast. Survey_of_mucoactive_use_in_UK_adult_critical_care_units_survey_of_p(.xlsx), PP_IMV_Mucoactive_agent_use_27082019(.xlsx), Unit_level_Mucoactive_agent_use_27082019(.xlsx). DOI: 10.17034/de3d8d3c-9c4b-459e-8b16-8fb9bb50e1d7.

## Supplemental Information

Supplemental information for this article can be found online at http://dx.doi.org/10.7717/peerj.8828#supplemental-information.

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
