# Peer review of "Mucoactive agent use in adult UK Critical Care Units: a survey of health care professionals’ perception, pharmacists’ description of practice, and point prevalence of mucoactive use in invasively mechanically ventilated patients"

_PeerJ, doi:10.7717/peerj.8828_

## Round 0.1 · original submission · Minor Revisions

The reviewers brought up a few relevant points/questions. Please address these as best as possible in the revision.

Reviewer 1 ·

Basic reporting

Basic reporting
Clear and easy to read language and form.
Introduction and rationale is clear with relevant literature.
Figures are clear.

Experimental design

Experimental Design
The study is reporting on 3 different surveys of clinical practice, perception of benefit and point prevalence use of mucoactive in ventilated patients. The design is appropriate to answer the aim of the work.

I would have like to see the exact questions that were asked in each survey and how phrased to ensure they were not leading - could the authors add this in supplementary documents.

I believe there is a review that should be referenced (Icard et al, Expert Rev Respir Med. 2017 Oct;11(10):807-814.). I do not think there are other meta-analysis or systematic reviews

Validity of the findings

Validity of the findings
This study reports the widespread use of mucoactives in ventilated patients in he UK. Although this is known anecdotally this is the first survey to show this and present the need for confirmatory studies to prove their efficacy or indeed harm in this patient group.

·

Basic reporting

no comment

Experimental design

no comment

Validity of the findings

The response rate to the practitioner level survey is quoted as 37% which is the proportion of ICUs who returned at least one reply. But the unit for analysis is the individual practitioner (n =255/3099) which suggests a lower response rate of 8%. This should be acknowledged in the text. It would also be useful to see if the 255 respondents were spread evenly across the units or if there were a concentration of replies from individual units. Likewise, reporting at least some results by role of respondent may be useful (of those who supported the idea of a clinical trial, how many were doctors/nurses/physiotherapists? What was the response rate in each of these groups)? At least one additional table showing the results in more detail would help – I would suggest two additional tables, one describing the characteristics of the responders (how many units, how many people, summaries of distribution of people across units, and counts and response rate by role) and one giving more detail of the responses by unit and role.

Additional comments

The results are presented in figs 1 to 4 using Borda counts, this should be explained in the methods section to aid interpretation by the reader.

---

## Round 0.2 · accepted · Accept

Thank you for addressing the reviewers' concerns. Congratulations again!